# Transgender Person and Pre-Exposure Prophylaxis for HIV: A Renal Dilemma

**Abu Baker Sheikh** [1,*] , **Nismat Javed** [2] and **Angela Dunn** [1]

1   Department of Internal Medicine, University of New Mexico, Albuquerque, NM 87131, USA; angela.dunn3@va.gov
2   Shifa College of Medicine, Shifa Tameer-e-Millat University, Islamabad 44000, Pakistan; nismatjaved@gmail.com
*   Correspondence: absheikh@salud.unm.edu; Tel.: +1-631-633-4125

**Abstract:** Sexual health care for transgender people is often inadequate and not addressed. Targeted prevention approaches that respond to the specific needs of transgender individuals are essential to reducing HIV infections. HIV prophylaxis is a proven intervention in the prevention of HIV among high-risk populations. However, creatinine clearance is one major determining factor in prescribing HIV pre-exposure prophylaxis. One of the variables used in the equation to calculate creatinine clearance is gender. Additionally, regarding transgender people, gender-affirming hormonal therapy also alters the clearance by modifying other variables, such as muscle mass. Here, we present the case of a 58-year-old designated female at birth, who transitioned to male 15 months ago, currently using testosterone, and had presented to the clinic requesting HIV pre-exposure prophylaxis, due to his anticipation of new sexual partners soon. He was initially denied HIV pre-exposure prophylaxis, due to lower estimated creatinine clearance when calculated per his natal assigned gender. The transgender population requires effective HIV pre-exposure prophylaxis, dependent on creatinine clearance, that is dictated by many factors, considering the high prevalence rate. Therefore, validation of eGFR equations in the transgender population is of utmost importance to ensure optimal decision-making and provision of health care.

**Keywords:** transgender; HIV; prophylaxis; creatinine

## 1. Introduction

Sexual health care for transgender people is often inadequate and not addressed. According to a 2017 estimate, about 1 million people in the US are transgender [1]. Lack of training and awareness on the part of the primary care provider has serious implications on the quality of care provided to the transgender community, particularly HIV prevention needs. Transgender men's sexual health is not well studied and the prevalence of HIV among transgender men might be under-reported. Transgender men, particularly those who have sex with cisgender men, are at high risk for contracting HIV infection [1]. Targeted prevention approaches, including HIV prophylaxis, that respond to the specific needs of individuals are essential to reducing HIV infections. HIV prophylaxis is a proven intervention in the prevention of HIV among high-risk populations. However, creatinine clearance is one major determining factor in prescribing HIV pre-exposure prophylaxis. One of the variables used in the equation to calculate creatinine clearance is gender.

## 2. Case Report

A 58-year-old designated female at birth with a past medical history of insomnia and depression who presented to clinic requesting HIV pre-exposure prophylaxis (PrEP), because he anticipate having new sexual partners soon. He is not currently sexually active. He is open to multiple types of sexual encounters, but has no planned partner and is not sure yet what sexual practices he will participate in. He achieved appropriate

developmental milestones as a child. The patient wanted to transition 16 years back but a lack of awareness of medical and surgical interventions available and resources held him back. He came through with his decision to transition 15 months ago with a good support system around him. The patient has no prior history of cerebrovascular accidents, thromboembolism, myocardial infarction, seizures, or fractures. He is currently using three pumps of topical 1.62% 20.25 mg testosterone per day and has been on testosterone for 15 months. He is tolerating it well without any side effects. He denies any alcohol use and smokes half a pack per day of cigarettes. Current medications apart from testosterone include vitamin D supplements, melatonin, folic acid, and magnesium oxide.

His hemoglobin was 14.7 g/dL and hematocrit was 44%. The patient's creatinine increased since the initiation of testosterone therapy. Creatinine before starting hormonal therapy was 0.9 mg/dL, and the recent laboratory findings showed creatinine of 1.2 mg/dL. Liver function tests were all within a normal range. The HIV antibody test, gonorrhea, chlamydia, and syphilis screen were negative. The urinalysis showed that the urine culture was negative and renal ultrasound showed normal size kidneys with no hydronephrosis.

Given the patient's age of 58, the weight of 147 pounds, and using the birth gender, the estimated creatinine clearance of 48 mL/min was calculated per Modification of Diet in Renal Disease (MDRD) and 52 mL/min per Chronic Kidney Disease Epidemiology Collaboration (CKD-EPI) creatinine equation. He was denied emtricitabine/tenofovir disoproxil fumarate (Truvada) PrEP for HIV prevention given that his estimated glomerular filtration rate (eGFR) was less than 60 mL/min. However, if his eGFR were calculated using his current gender, which turned out to be greater than 60 mL/min, he would have qualified for the PrEP (Table 1).

**Table 1.** Estimated eGFR according to gender for both equations.

| Equation | Natal Gender (Female) | Current Gender (Male) |
|---|---|---|
| CKD-EPI for eGFR (mL/min/1.73 m$^2$) | 52.0 | 69.0 |
| MDRD for eGFR (mL/min/1.73 m$^2$) | 48.0 | 66.1 |
| Cockcroft-Gault equation (mL/min) | 54.0 | 63.0 |

CKD-EPI—Chronic Kidney Disease Epidemiology Collaboration, MDRD—Modification of Diet in Renal Disease, eGFR—estimated glomerular filtration rate.

## 3. Discussion

HIV is progressively becoming a dangerous and prevalent disease. According to estimates from a survey in 2018, about 1.2 million people aged 13 and older had HIV in the United States [2]. This situation becomes complicated when a particularly high-risk population—due to social, legal exclusion, economic vulnerability, discrimination, and marginalization—the transgender population, is brought into consideration. From 2009 to 2014, about 2300 transgender people were diagnosed with HIV, of whom 84% were transgender women and 15% were transgender males [1]. These factors place the patient in question at high risk of contracting HIV.

One of the ways of preventing the infection is the usage of pre-exposure prophylaxis. Currently, oral tenofovir disoproxil fumarate (TDF)-FTC, also known as Truvada, and oral tenofovir alafenamide-emtricitabine (TAF)-FTC, also known as Descovy, are the only FDA-approved regimen for HIV PrEP. Truvada and Descovy are indicated in persons with an estimated creatinine clearance greater than 60 mL/min and 30 mL/min, respectively [3]. However, the dosage of the drug is dependent upon creatinine clearance. Our patient was not a candidate for Descovy given that the findings of the DISCOVER trial, which compared both TAF-FTC and TDF-FTC for PrEP, only evaluated individuals who engaged primarily in anal-receptive sex. The results cannot be generalized to those who engage in

vaginal sex, and it is generally avoided in women and transgender men [4]. Although the drug itself is not known to cause massive variations in the clearance, several factors could manipulate the clearance, leading to problems in dosing [5].

The Modification of Diet in Renal Disease (MDRD) Study equation and the Chronic Kidney Disease Epidemiology Collaboration (CKD-EPI) equation are the most widely used methods to calculate eGFR [6]. Their use is not validated in the transgender population. It is unclear when using eGFR equations for a transgender person whether to use natal assigned sex or current gender after the transition. The patient was currently using masculinizing therapy for the development of masculine secondary sexual characteristics that match the identity. This leads to many physiological changes such as the deepening of the voice, menstrual suppression, facial and body hair growth, increased muscle mass, and body weight [7]. Klaver et al. reported that the use of testosterone therapy increased lean mass by 3.9 kg in transgender men [8]. Therefore, the change in creatinine, in theory, could impact clearance calculation; but whether there is a statistical difference in actual change in GFR is currently not studied.

There is no literature suggesting the ranges that should be used when faced with such a dilemma, such as this case, apart from one recent article that discusses the approach to estimated glomerular filtration rate in the transgender population [9]. It has been suggested that both male and female sexes should be considered in eGFR equations for transgender persons on gender-affirming hormone therapy that would provide a range of eGFR by sex that can be narrowed by deciding on which value likely reflects the muscle mass of the patient to which the calculations were applied. That value of muscle mass can then be used for dosing medications [9].

## 4. Conclusions

The transgender population requires timely and effective HIV pre-exposure prophylaxis considering the high prevalence rate. HIV PrEP is dependent on creatinine clearance that can be dictated by many factors. Overestimation and underestimation of eGFR can both lead to disastrous outcomes. Therefore, validation of eGFR equations is needed in the transgender population to improve clinical decision-making and prevent delays in accessing care.

**Author Contributions:** A.B.S. and N.J. were involved in the conception, data interpretation, literature review, and writing of the manuscript. A.D. was involved in the conception, writing, and critical review of the manuscript. All authors have read and agreed to the published version of the manuscript.

**Funding:** This research received no external funding.

**Institutional Review Board Statement:** The study was conducted according to the guidelines of the Declaration of Helsinki and presented fully anonymized non-identifiable information of a single-patient case report that did not require prior approval by the Institutional Review Board.

**Informed Consent Statement:** Written informed consent was obtained from the patient involved in the study.

**Data Availability Statement:** Data may be made available on reasonable request from the corresponding author.

**Conflicts of Interest:** The authors declare no conflict of interest.

**Ethics/Patient Consent:** Written informed consent was obtained from the patient prior for this report.

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
