# Peer review of "Transgender Person and Pre-Exposure Prophylaxis for HIV: A Renal Dilemma"

_2036-7449, doi:10.3390/idr13020043_

Round 1
Reviewer 1 Report
In this paper by Sheikh et al, narrate a case report, where a transgender, now male, was initially denied HIV pre-exposure prophylaxis due to lower estimated creatinine clearance when calculated per his natal assigned gender. Their point is changing gender from female to male, parameters for glomerular filtration rate (eGFR) should be adjusted by keeping into consideration the gender change.
Comments:
It is an important issue but is not the do-or-die issue. If explained well to the doctor, sometimes, may require taking advice from more than one doctor, the problem can surely be taken care off. The authors should consider that sex change brings other physical changes in the individual, and as authors suggested are not yet studied or experienced sufficiently. Thus, in order to make this observation or case study significant, the authors need to mention other cases as well, and how different doctors solved similar problems with those circumstances/transgenders.
Minor Comments
It is a very well-written narration of the subject. However, the manuscript needs English editing at some places, as there are mistakes that require rectification before publication.
Author Response
Dear Respectful Reviewer,
Thank you for providing your valuable feedback, it has helped us in improving our manuscript.
Reviewer Comment 1 - It is an important issue but is not the do-or-die issue. If explained well to the doctor, sometimes, may require taking advice from more than one doctor, the problem can surely be taken care off. The authors should consider that sex change brings other physical changes in the individual, and as authors suggested are not yet studied or experienced sufficiently. Thus, in order to make this observation or case study significant, the authors need to mention other cases as well, and how different doctors solved similar problems with those circumstances/transgenders.
Response- This case report is an attempt to highlight this unique issue in transgender people which is not been previously reported. We want to engage the doctor community and open the floor for discussion worldwide through this case report. Also, the previous studies discuss the impact of pre-exposure HIV prophylaxis on many parameters, but the pre-requisites required for individuals to be facilitated for the prophylaxis depends on many factors, especially creatinine clearance, as was the case for the patient. Additionally, gender-affirming therapies also change the clearance, further causing difficulties in the prescription of prophylaxis both points have been highlighted in our case report.
Reviewer Comment 2- It is a very well-written narration of the subject. However, the manuscript needs English editing at some places, as there are mistakes that require rectification before publication.
Response – The article has been revised to omit grammar errors, space errors and article usage errors.
Reviewer 2 Report
The case report highlights a very important issue with the transgender community where their sexual health is not addressed adequately. The authors have provided with sufficient background information and in their case report also give necessary health information and prove that the eGFR according to the current accepted equation is different for natal gender and current gender of a transgender person which might hinder there qualification for the PrEP. Authors give a vital suggestion of validating eGFR equation for the transgender community which would prevent any delays in their access to PrEP.
The authors define MDRA and CKD-EPI in lines 93 and 94 but use it for the first time in lines 63 and 64, so it would be better to define it the first time the acronyms are used.
Author Response
Dear Respectful Reviewer,
Thank you for providing your valuable feedback, it has helped us in improving our manuscript.
Reviewer Comment 1: The authors define MDRD and CKD-EPI in lines 93 and 94 but use it for the first time in lines 63 and 64, so it would be better to define it the first time the acronyms are used.
Response: The acronyms have been defined in lines 63 and 64 as “Modification of Diet in Renal Disease (MDRD)” and “Chronic Kidney Disease Epidemiology Collaboration (CKD-EPI)”.
Round 2
Reviewer 1 Report
it is fine now